# Estimating Crop Sowing and Harvesting Dates Using Satellite Vegetation Index: A Comparative Analysis

**Grazieli Rodigheri** [1,2,*], **Ieda Del'Arco Sanches** [1,3], **Jonathan Richetti** [2], **Rodrigo Yoiti Tsukahara** [4], **Roger Lawes** [2], **Hugo do Nascimento Bendini** [3] **and Marcos Adami** [1,3]

1   Remote Sensing Postgraduate Program (PGSER), Coordination of Teaching, Research and Extension (COEPE), National Institute for Space Research (INPE), Av. dos Astronautas, 1.758, São José dos Campos 12227-010, SP, Brazil; ieda.sanches@inpe.br (I.D.S.); marcos.adami@inpe.br (M.A.)
2   CSIRO, 147 Underwood Av., Floreat, WA 6014, Australia; jonathan.richetti@csiro.au (J.R.); roger.lawes@csiro.au (R.L.)
3   Earth Observation and Geoinformatics Division (DIOTG), General Coordination of Earth Science (CG-CT), National Institute for Space Research (INPE), Av. dos Astronautas, 1.758, São José dos Campos 12227-010, SP, Brazil; hugo.bendini@inpe.br
4   Fundação ABC Pesquisa e Desenvolvimento Agropecuário, Rua Jonas Borges Martins, 1313, Castro 84165-250, PR, Brazil; rodrigo@fundacaoabc.org
*   Correspondence: grazieli.rodigheri@inpe.br; Tel.: +55-54-999912470

**Abstract:** In the last decades, several methodologies for estimating crop phenology based on remote sensing data have been developed and used to create different algorithms. Although many studies have been conducted to evaluate the different methodologies, a comprehensive understanding of the potential of the different current algorithms to detect changes in the growing season is still lacking, especially in large regions and with more than one crop per season. Therefore, this work aimed to evaluate different phenological metrics extraction methodologies. Using data from over 1500 fields distributed across Brazil's central area, six algorithms, including CropPhenology, Digital Earth Australia tools package (DEA), greenbrown, phenex, phenofit, and TIMESAT, to extract soybean crop phenology were applied. To understand how robust the algorithms are to different input sources, the NDVI and EVI2 time series derived from MODIS products (MOD13Q1 and MOD09Q1) and from Sentinel-2 satellites were used to estimate the sowing date (SD) and harvest date (HD) in each field. The algorithms produced significantly different phenological date estimates, with Spearman's R ranging between 0.26 and 0.82 when comparing sowing and harvesting dates. The best estimates were obtained using TIMESAT and phenex for SD and HD, respectively, with R greater than 0.7 and RMSE of 16–17 days. The DEA tools and greenbrown packages showed higher sensitivity when using different data sources. Double cropping is an added challenge, with no method adequately identifying it.

**Keywords:** time series; Google Earth Engine; remote sensing; soybean; phenological metrics

## 1. Introduction

Remote sensing has been widely used to characterize the seasonal dynamics of vegetation at continental and global scales [1]. In the last two decades, many studies have been conducted at different scales and provided a set of techniques and algorithms to extract phenological information from orbital data, defined as Land Surface Phenology (LSP) [2–5]. The main advantage is that these metrics are a crucial component in crop modeling simulations that determine how weather, disease, and pests will affect crop yields [6–15] and can also be used to plan to harvest to minimize losses [9]. Although various methods have been developed, accurately identifying phenological events over large areas is still challenging [16] as it demands a high computing time [17]. Thus, evaluating the different methods and their sensitivity to different data sources is warranted.

To extract sowing and harvest dates from earth observation, several time-series methods have been developed [11,16,18–21]. The first phenology extraction software is the TIMESAT, released in the early 2000s [2,22,23]. Since then, many researchers have sought to implement improvements in extracting vegetation phenology. Many have continued to implement methodologies for LSP extraction, such as CropPhenology [18], phenex [24], phenofit [21], greenbrown [25], rTIMESAT [26] developed in R language, and Digital Earth Australia tools (DEA) [27] in Python. Although much effort has been spent on developing and improving different phenology extraction packages, little effort has been conducted to evaluate and compare the differences between them. Furthermore, there is a gap in understanding the robustness of such algorithms when input with different data sources, such as data from different satellite platforms and their products.

Phenology estimation from earth observation is challenging because different platforms provide different remote sensing products with issues to be dealt with, such as noisy data from cloud cover and low temporal, spatial, or radiometric resolution [28–30]. The most common remote sensing data for LSP is the Moderate Resolution Imaging Spectroradiometer (MODIS) sensor on board the Terra and Aqua satellites, which provides consistent spatial–temporal data [22,31,32]. However, with the wide availability and reliability of newer platforms, finer spatial resolution can present more information about vegetation seasonality and possibly more adequate spatial resolutions to investigate changes in LSP [33,34]. Thus, an assessment of the impact of different data sources to assess LSP is needed to evaluate this method's robustness.

In addition, from a practical point of view, a significant obstacle in obtaining satellite phenology is the incompatibility between ground-based and satellite-based observations [35]. The main cause of this incompatibility is the inherent difference in their respective definitions of phenology [28]. That is, the start of the season is estimated at the time when green leaf reflections are detected by the sensors [36], and it is associated with the emergence date rather than the sowing date of the crop. However, it is often used interchangeably as a sowing date [37], and the lag between the actual sowing date and plant emergence [17] is not natively considered in LSP algorithms, making the process of extracting these dates more difficult. In addition, the lag and uncertainties are even greater for pixels with coarser resolutions. The lag increases since SOS is generally detected later in pixels with coarser resolution [1], while uncertainties increase due to the spectral mixing of wider pixels, generating irregular curves [38] that can affect estimates. Thus, the comparison of the different algorithms needs to account for the impact of the different data sources in order to develop a more general understanding of which method(s), platform(s), or product(s) can be used to accurately map sowing and harvest dates for large agricultural areas.

Therefore, we sought to evaluate and compare different algorithms for extracting sowing and harvest dates in large regions (1568 fields) and their performance when inputting different data sources. The tested algorithms included CropPhenology, Digital Earth Australia tools package (DEA), greenbrown, phenex, phenofit, and TIMESAT. The impact of contrasting data sources was evaluated using data retrieved from MODIS and MultiSpectral Instrument (MSI)/Sentinel-2 sensors. To our best knowledge, this is the first study that aimed to compare the most used algorithms developed in the last two decades, inputting multiple data sources.

## 2. Materials and Methods

### 2.1. Study Area

This study's area spreads across four Brazilian states, namely, Paraná (PR), São Paulo (SP), Goiás (GO), and Minas Gerais (MG). A total of 1568 fields (Figure 1a) were kindly provided by the ABC Foundation "https://fundacaoabc.org/ (accessed on 15 July 2023)", with observed soybean sowing and harvest dates during the 2019–2020 growing season. The samples represent soybean farms monitored by the ABC Foundation, where observations were conducted by the landowner and reported to the institution.

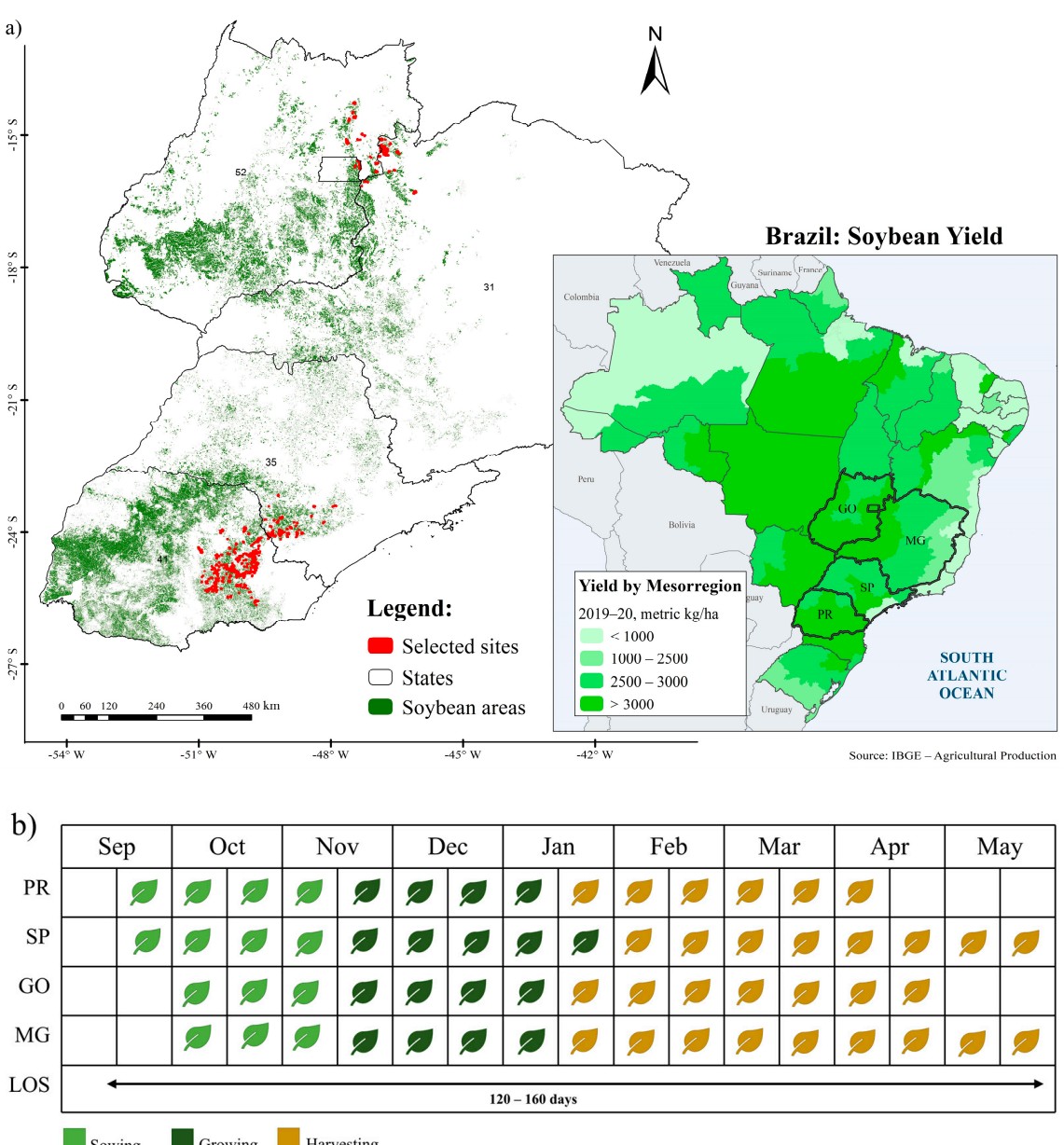

**Figure 1.** Overview of this study's area showing the location of the phenology observation fields used for validation and total soybean cropped areas (**a**). Usually, the soybean calendar and length of the season (LOS) for the states of Paraná (PR), São Paulo (SP), Goiás (GO), and Minas Gerais (MG) study area vary from 120 to 160 days (**b**). Soybean areas downloaded from: [39].

In this study's area (Figure 1a), the soybean growing season is modulated by rainfall. In Paraná, soybean sowing usually occurs between the end of September (when the rainy season starts) and mid-November, with a higher concentration between late October and early November [37,40,41]. Harvesting is usually performed between mid-January and early April, with the crop cycle in most regions ranging from 120 to 160 days [37,40,41]. In São Paulo, the sowing dates and the crop cycle duration are quite similar to those performed in Paraná, while the harvest comprises the period from February to May (Figure 1b; [42]. In Goiás and Minas Gerais, sowing is a little later and can occur until mid-December. The harvest can extend from January to April in the state of Goiás and until May in Minas Gerais [42]. Thus, in order to capture the growing season of all regions, our study period was extended from the end of August 2019 to the beginning of May 2020.

## 2.2. Remote Sensing Data

To understand how robust the algorithms are to varying inputs, data from two different data sources, MODIS (products MOD09Q1, MOD13Q1, MYD13Q1) and MSI /Sentinel-2 (Level—2A) that were available in Google Earth Engine (GEE) were used. Normalized difference vegetation index (NDVI) [43] and enhanced vegetation index (EVI) [44] have been the most commonly used vegetation indices in retrieving vegetation phenology [45]. However, the estimates vary among methods [4] and the crop season [45], with very different or even conflicting results [4,45]. Thus, to have a comprehensive evaluation of how the different indices can affect the algorithms, we used the NDVI and two-band enhanced vegetation index (EVI2) [46]. The MOD13Q1 and MYD13Q1 products offer a 16-day maximum value composition (MVC) of NDVI and EVI2 indices at a spatial resolution of 250 m. Aggregated, the MOD and MYD products can offer an 8-day temporal resolution. The MOD09Q1 product offers an 8-day MVC in the red and near-infrared regions of the electromagnetic spectrum at 250 m, while the MSI/Sentinel-2 sensors collect data every five days at a resolution of 10 m in similar spectral regions. For the reflectance data, NDVI and EVI2 for each field pixel in each image were calculated according to Equations (1) and (2) [43,46].

$$NDVI = NIR - Red/NIR + Red \qquad (1)$$

$$EVI2 = 2.5 \times (NIR - Red)/(NIR + (2.4 \times Red) + 1) \qquad (2)$$

where NIR and Red are near-infrared and red reflectances; both vegetation indices were used since most of these packages have not been tested before using different indices. Thus, the algorithms were assessed with varying temporal, spatial, radiometric, and spectral resolutions, totaling 48 different experiments (6 packages × 4 data sources × 2 indexes).

## 2.3. Satellite Data Pre-Processing

Because the spectral reflectance value of each pixel is partially influenced by the spectral properties of adjacent pixels [47], pixels at the field border were excluded using a negative buffer of 125 m, as described below. The datasets were then processed with the GEE platform [48] to remove the noise data, such as pixels affected by clouds and cloud shadows (Figure 2). Then, after removing the noise, the whole field NDVI and EVI2 time series were generated. The NDVI and EVI2 values were averaged to provide the whole field value. Product-specific pre-processing steps were also applied, as described in the following sections.

### 2.3.1. MOD13Q1/MYD13Q1 Processing

The MOD13Q1 and MYD13Q1 product's Quality Assessment (QA) reliability was used to identify and remove noisy pixels. To accomplish this, a mask was applied to each image, selecting data only with QA = 0 (good quality) and QA = 1 (marginal data). We also selected pixels where the blue reflectance band value was less than 10%. To reduce the potential effect of anisotropic reflectance, we only used pixels with a view zenith angle of less than 30°. After the corrections, the MOD13Q1 and MYD13Q1 products were combined to generate an 8-day product, referred to as MCD13.

### 2.3.2. MOD09Q1 Processing

For MOD09Q1, the quality band was used to filter out the disturbance that could affect the Vegetation Index (VI) data quality. To do this, a mask was created using the State band and filtering the bits 0 and 1 (Cloud), 2 (Cloud shadow), and 8 and 9 (Cirrus cloud).

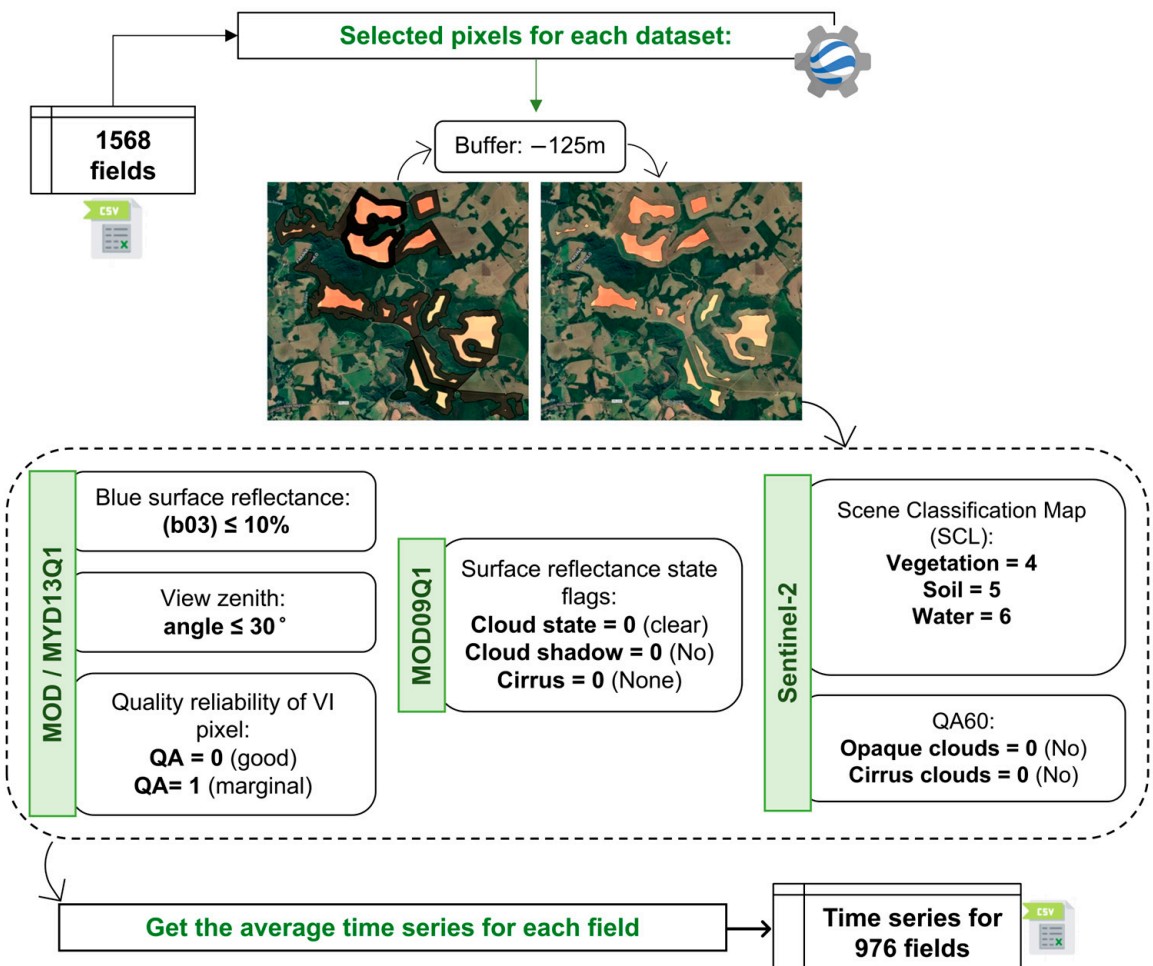

**Figure 2.** Flowchart of the NDVI and EVI2 time series extraction process on the Google Earth Engine platform.

### 2.3.3. Sentinel-2 Processing

For Sentinel-2 (S2), the data containing clouds or noisy data were filtered using the Scene Classification Map (SCL) band provided with the dataset. To do this, all pixels classified with values 1 (Saturated), 2 (Dark), 3 (Cloud shadow), 7 (Cloud_low), 8 (Cloud_medium), 9 (Cloud_high), 10 (Cirrus), and 11 (Snow/Ice) were removed. In addition, the cloud mask band (QA60) provided with the dataset was used to remove opaque and cirrus cloud noise.

### 2.4. Sowing and Harvest Dates Estimation

The phenological attributes from VI datasets were obtained in two steps (Figure 3): (1) Time-series smoothing (see Supplementary Materials—Section S1); and (2) Parameter calibration (see Supplementary Materials—Section S2). After this, the validation and analysis of the phenological metrics were performed. In the validation phase, the influence and differences caused in the estimates by the different packages and data sources were analyzed. Also, because all methods provide Start of Season (SOS) and End of Season (EOS), the sowing dates were defined as SOS minus 10 days (see Section 2.6), and harvest dates were defined as the EOS date.

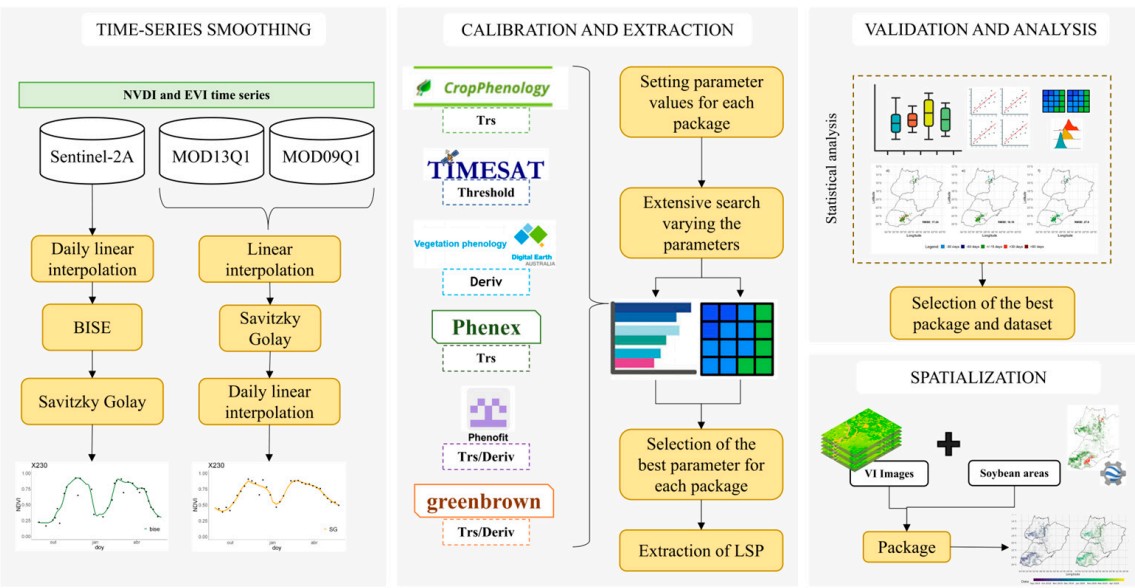

**Figure 3.** Flowchart of the three stages of extraction of phenological metrics.

### 2.4.1. Time-Series Smoothing

Even though most of the noise data were removed during the pre-processing steps, the VI time series required further steps such as filtering [3]. Different methodologies were used for each dataset after evaluating various noise removal methods (see Supplementary Materials—Section S1, Figures S1 and S2). Briefly, for MODIS data that already had the maximum-value composite procedure (MVC), each time series was interpolated to fill existing gaps, filtered with Savitzky–Golay (SG) [3] to remove residuals, and interpolated for daily values. Each time series in the Sentinel-2 dataset was first interpolated daily to fill in gaps and null data (days without images). The Best Index Slope Extraction (BISE) algorithm [49] was then used to remove the remaining noise/outliers from the set of time series. Subsequently, the SG filtering procedure was used to smooth the time series VI curve and reduce residual noise in the time series VI datasets. The SG filter was applied to all datasets with a window size of 5 and a polynomial degree of 2.

As each extraction algorithm had different functions for noise removal and time series smoothing, this processing was performed separately before LSP extraction in order to provide the algorithms with the best time series for each product. These processes reflect standard procedures when analyzing VI time series [3].

### 2.4.2. Phenology Extraction Algorithms

a.    CropPhenology

The CropPhenology (CP) package focuses only on phenology metric extraction, and the function extracts consecutive VI values in the time series of images for each pixel to define a space–time cube, a three-dimensional dataset that represents the time series of VI values (latitude, longitude, and time) [18]. The package provides the user with 15 phenological metrics that represent the seasonal growth condition of the crop for each pixel [18]. In this package, the estimates of the start and the end of the season were implemented by starting at peak development and analyzing the VI changes between the previous and later values, iteratively looking for local minima (dips) in VI values. Therefore, the SOS and EOS are each defined as the first minimum below the user-specified VI threshold and the last minimum below the threshold value, respectively [18].

b.    Digital Earth Australia tools package

The Digital Earth Australia (DEA) function xr_phenology can calculate several LSP statistics that together describe the characteristics of a plant's lifecycle. This function is

contained in the dea_tools.temporal script that is a part of the DEA big data platform that provides both the analytical tools and the high-performance computing infrastructure needed to facilitate the analysis of multi-decadal time series data [50]. The package was developed in Python and had two possible parameter values for estimating each of the SOS and EOS metrics, as follows [27]: If 'first', then SOS is estimated as the first positive slope on the greening side of the curve. If 'median', then SOS is estimated as the median value of the curve's positive slopes on the greening side. If 'last', then EOS is estimated as the last negative slope on the senescing side of the curve. If 'median', then EOS is estimated as the 'median' value of the negative slopes on the senescing side of the curve.

c.  Greenbrown

The greenbrown package (GB) is a collection of functions developed to analyze trends, trend changes, and phenology events from satellite observations or climate model simulations [25]. In this package, phenology events can be estimated from the daily interpolated, gap-filled, and smoothed time series using two different methodologies, thresholds (PhenoTrs and PhenoWhite), or extreme values of the derivative of the seasonal cycle (PhenoDeriv) [25]. Both approaches are based on the definition of SOS and EOS as the mid-points of green-up and senescence, respectively [51].

d.  Phenofit

The phenofit (PF) package was developed in the R language and had several built-in functionalities, from noise minimization to metric extraction [21]. The input supports evenly and unequally spaced time series and can handle missing input values. It combines methods of growing season splitting, 7 curve fitting methods, and four phenology extraction methods and provides flexible input and output routines [21]. In this package, the threshold and derivative method were also selected, which defines SOS and EOS as the first day on which the VI value exceeds the threshold and mid-points of green-up and senescence, respectively.

e.  Phenex

The phenex (PX) package is implemented in the R language and provides a collection of functions for the temporal analysis of phenological data [24]. Besides having methods for extracting phenological metrics, the package has functions such as BISE [49] for correcting time series and methods such as linear interpolation, fast Fourier transform (FFT), and SG filter for fitting time series [24]. The function modelValues allows for these corrections to be performed while the function phenoPhase extracts the phenological metrics from the time series based on the threshold method [24]. This function has as parameters the time series, the desired phase to be extracted (e.g., green-up, max, or senescence), the threshold method (local or global), and the threshold value (e.g., 0.20, 0.50) [24].

f.  TIMESAT

The TIMESAT (TM) is a software package for analyzing time series of satellite sensor data [2]. The TIMESAT program [52] provides eleven phenological metrics, including the start and end dates of the growing season. The TIMESAT software provides three different approaches for smoothing the VI data and fitting it to a chosen function [52]. In this package, a season officially starts when the value has increased by a specific measure (threshold) of the distance between the minimum level on the left and the maximum level, as determined by the filtered or modified functions. Similar definitions are given for the end of the season [52]. In this work, the rTIMESAT was used [26]. The package requires at least two annual cycles to run and returns all identified seasons. Thus, in this paper, we double the time series by extending its size twice.

*2.5. Parameter Calibration and Extraction*

Before extracting phenological metrics, a calibration was carried out using 200 randomly sampled fields to determine the best configuration of each package for this study's

region (see Supplementary Materials—Section S2). The parameters that showed the highest correlation and fit with the field data were selected (Table 1). However, for the GB and PF packages, only the derivative method was used, as they showed better results. Thus, the evaluation was focused on three algorithms with the threshold method (CP, PX, and TM) and 3 with the derivative method (DT, GB, and PF), as they have shown consistent results in previous work [53–56].

**Table 1.** Parameters of each package used for validation of the phenological metrics, with CP (CropPhenology), DT (DEA tools), PX (phenex), and TM (TIMESAT).

| Package (Parameters) | CP (Threshold) | DT (Derivative) | PX (Threshold) | TM (Threshold) |
|---|---|---|---|---|
| MCD13 | (0.25, 0.35) | (median, last) | (0.15, 0.15) | (0.15, 0.20) |
| MOD09 | (0.30, 0.35) | (median, last) | (0.15, 0.15) | (0.15, 0.20) |
| MOD13 | (0.25, 0.25) | (median, last) | (0.15, 0.15) | (0.15, 0.20) |
| Sentinel-2 | (0.10, 0.40) | (median, last) | (0.10, 0.10) | (0.10, 0.10) |

Numbers and methods in parentheses indicate the parameter value for the sowing and harvest dates, respectively.

### 2.6. Performance Evaluation and Spatialization

Because farmers report sowing and harvest day and not the emergence or start and end of seasons, to estimate the sowing date, 10 days were subtracted from the detected SOS (provided by all algorithms that represented plant emergence) dates because this is the average time when soybean plants emerge and become visible to satellites [36,57,58]. Thus, it can be assumed that the gaps found between the estimates are due to the error between the data estimated by remote sensing and the data measured in the field. Shapiro–Wilk normality test [59], Spearman's rank correlation coefficient (R), the Root Mean Square Error (RMSE), bias (%), and linear regression between the reference field data and the estimates were used to assess the performance of the algorithms. The non-parametric Wilcoxon paired-samples test [60] (also known as the Wilcoxon signed-rank test) was used to assess whether there was a statistically significant difference between the estimates and the field data. The processing times (seconds) were also recorded.

Lastly, after the validation step, a map of the soybean season's sowing date, harvest date, and length was generated using one of the packages for the entire region (four states combined) for the 2020–2021 season. The mask with soybean areas for the 2020–2021 season is available in [39].

## 3. Results

The best estimate for SD was obtained using TM with MCD13 data source and EVI index, with RMSE of 16 days, R of 0.8, and bias of 10.4%. The best estimate for HD was obtained using PX with S2 data source and EVI index, with RMSE of 18 days, R of 0.75, and bias of –0.2%. Both packages showed distributions similar to the field data distribution (Figure S5). DT and GB showed the worst results, with most correlations below 0.5 and errors greater than 40 days (Table S1). The CP and PF packages showed low errors when compared to the field data but were less correlated; in general, R ranged between 0.5 and 0.7. Although some results have shown greater similarity with the ground-based data, it is still possible to observe many uncertainties in the estimates (the number of outliers in Figure 4 and dispersion around the 1:1 line in Figure 5).

Although performance differences exist when changing data sources, some algorithms perform similarly when changing platforms (Table S1 and Figure 4). For example, TM's biggest difference in R was less than 0.1 from MCD13 EVI2 to MOD09 NDVI. However, DT and GB were the most sensitive to data source changes at harvest, with DT's R varying from 0.29 with MOD13 NDVI to 0.47 on Sentinel-2 NDVI and at sowing with GB's R from 0.25 to 0.51 from MCD13 NDVI to MOD13 NDVI. Overall, EVI2 showed better results and more correlation with field data for all packages (Table S1 and Figure 4). The SD estimates were statistically equal to the data observed in the field using the CP and PF package and

MODIS and Sentinel-2 datasets, as well as the PX package using the MODIS dataset and TM using the MOD13 dataset (Figure 4). For the HD estimates, the CP and PX packages showed similarity to field data using MODIS and Sentinel-2 datasets, as well as the PF package using MODIS data. Only EVI2 was used in subsequent evaluations as it showed similar and slightly better results to NDVI.

Although most of the fields estimated by the TM package achieved a good fit, some fields could not be estimated adequately, especially the very low values that were associated with the package not returning estimates for the time series. For example, null values for SD or high values for HD indicated that the package ran through the time series and picked the end of the series without finding the end of the crop cycle (Figure 5).

There was a difference in the estimates between the different regions. In general, the estimates were lower than the observed data for the northern region and higher in the southern region (Figure 6 and S6–S9). For both estimates, most of the packages were consistent with the field observation, except for the DT and GB packages.

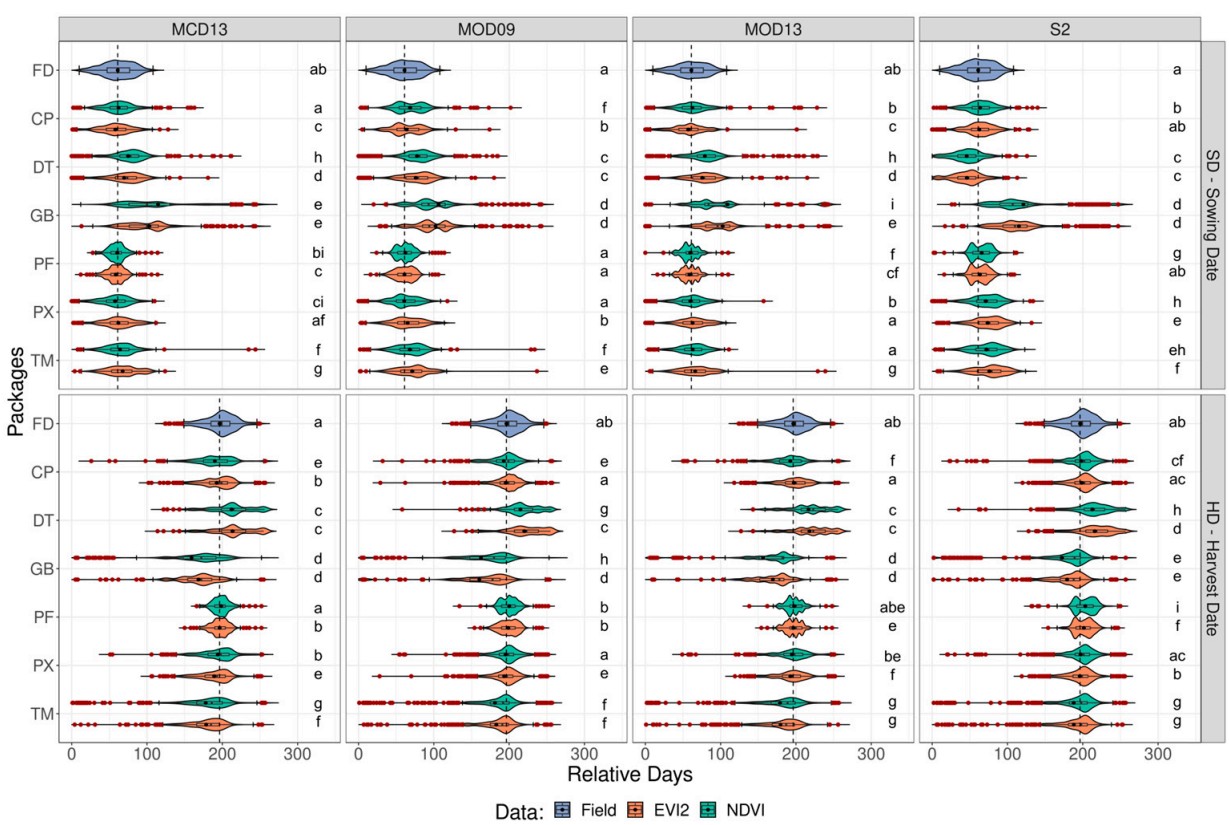

**Figure 4.** Violin plot comparing field data (FD) with sowing (SD) and harvest (HD) dates estimates for the different packages, CP (CropPhenology), DT (DEA tools), GB (greenbrown), PF (phenofit), PX (phenex), and TM (TIMESAT), using different data sources, MODIS (MCD13, MOD09, MOD13) and Sentinel-2 (S2).

TM was the most robust algorithm with more consistent results regarding RMSE, bias, and R (Table S1), regardless of changes in the data source. The GB package, contrarily, was the most sensitive to changes in the data source. However, all methods have some sensitivity to changes in data sources. For example, the less sensitive (TM) for SD has an average error between Sentinel-2 and MOD13 of about 10 days (Figure 7a) while almost 15 days from Sentinel-2 to MCD13 (Figure 7b). In the most sensitive method (GB), the average errors are more than 15 days apart for both SD and HD (Figure 7c,d).

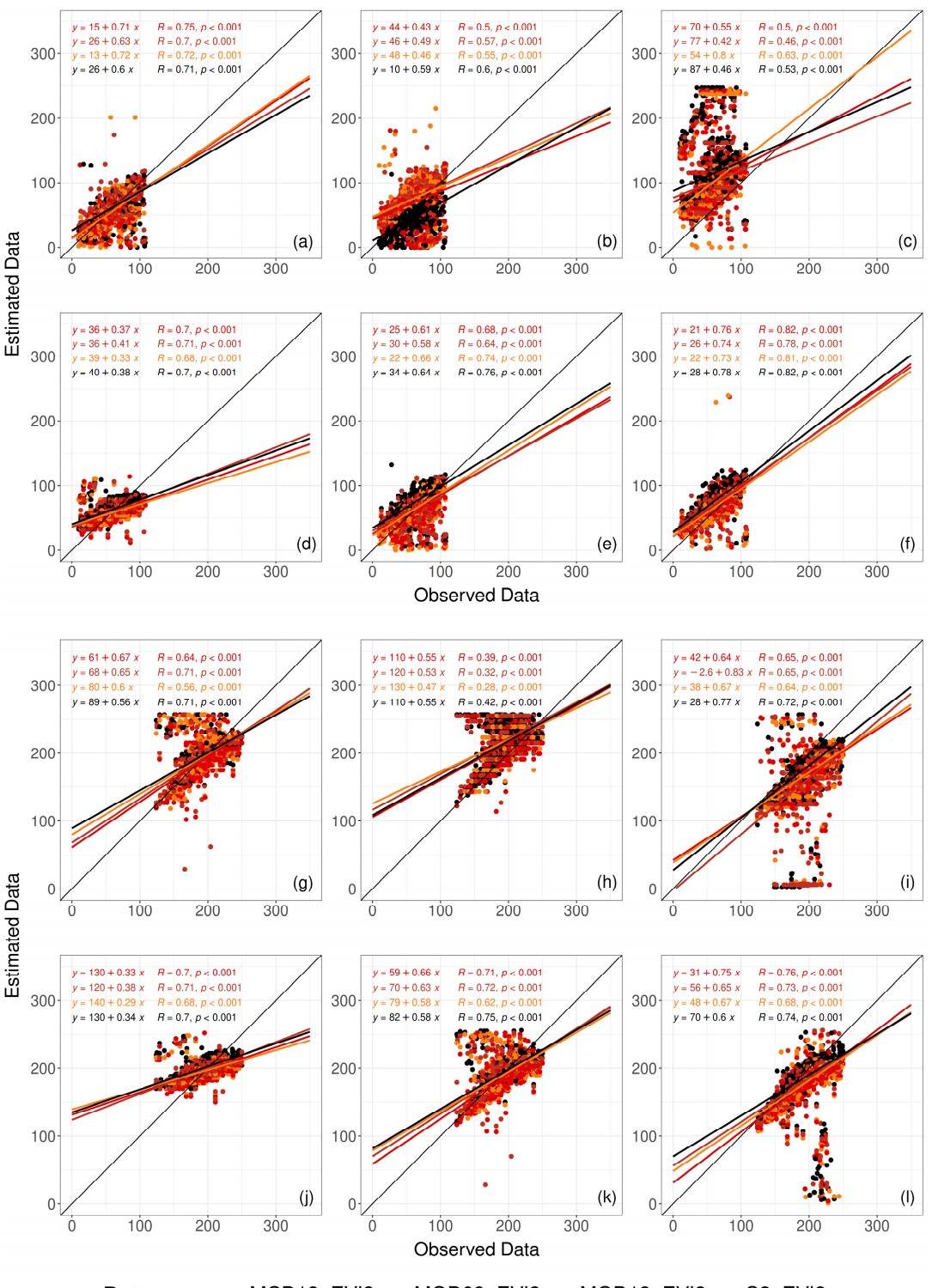

**Figure 5.** Dispersion between the data observed in the field and sowing (**a**–**f**) and harvest (**g**–**l**) estimates for the different datasets and packages tested, CropPhenology (**a**,**g**), DEA tools (**b**,**h**), greenbrown (**c**,**i**), phenofit (**d**,**j**), phenex (**e**,**k**), and TIMESAT (**f**,**l**). The black line represents the 1:1 line.

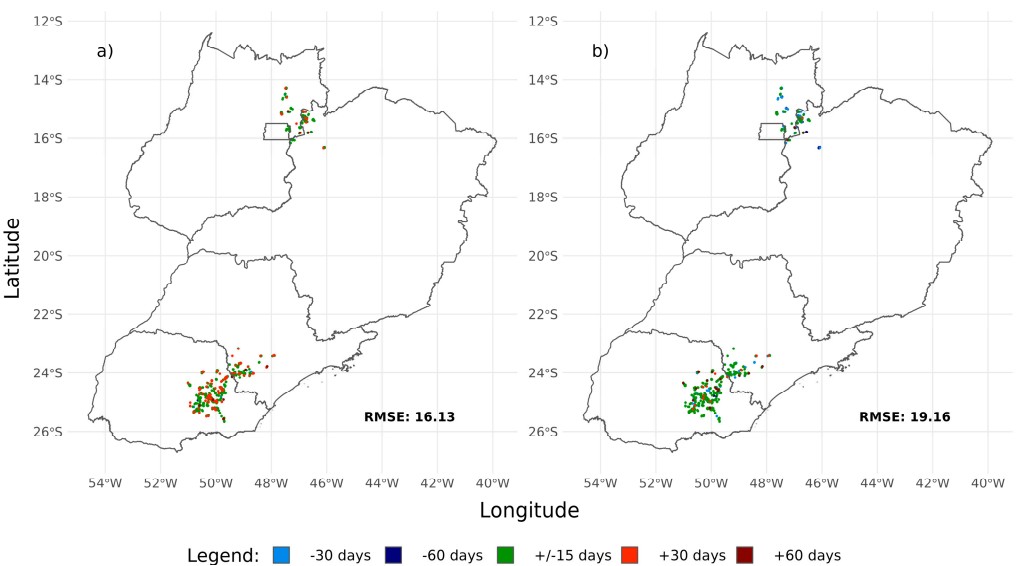

**Figure 6.** Spatial pattern of the difference between observed and estimated sowing date for TIMESAT (**a**) and observed and estimated harvest date for phenex (**b**) in days. The figures consider estimates for the MCD13Q1 EVI2 data source. Root Mean Square Error (RMSE) is in days.

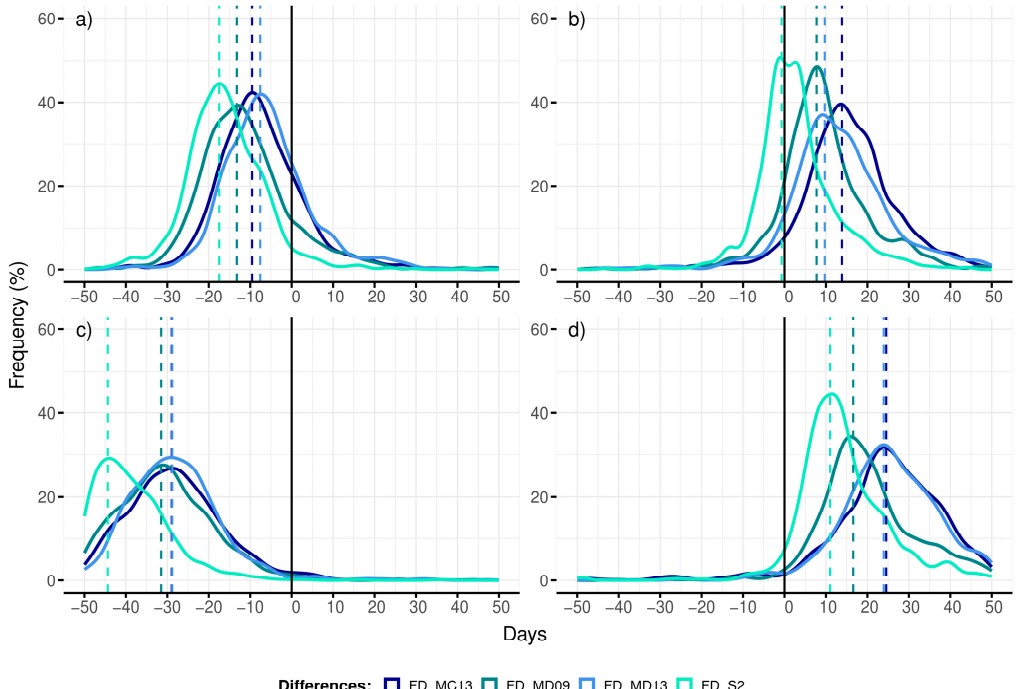

**Figure 7.** Distribution of the difference between the field data and sowing (**a**,**c**) and harvest estimates (**b**,**d**) using the different data sources, EVI index, and TM (**a**,**b**) and GB (**c**,**d**) packages. FD_MC13: difference between field data and MCD13; FD_MD09: difference between field data and MOD09; FD_MD13: difference between field data and MOD13; FD_S2: difference between field data and Sentinel-2. Dotted lines represent the peak of each distribution.

There are inconsistencies in some estimates made by the packages. The major source of errors is double cropping (lower R and more outliers). Packages assume only one crop in the mentioned season, often picking the wrong sowing or harvest date. In this case, even though TIMESAT has features to determine the number of cycles, it often fails to identify multiple cycles, particularly for harvest estimation (Figure 8).

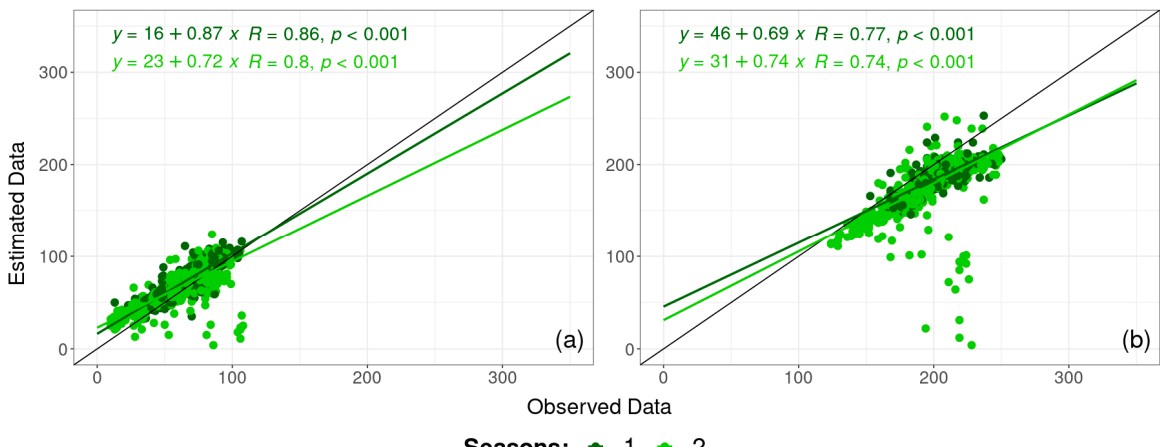

**Figure 8.** Dispersion between observed field data and sowing (**a**) and harvest (**b**) estimates for different numbers of seasons. Only the TM package and MCD13 data source were used as they showed the best fit (see Figure 5). The black line represents the 1:1 line.

The main goal of these algorithms is to quickly produce a sowing and harvest date map over a large area. Even though the algorithms are based on point scale, some of them provide features that enable the input of rasters, which facilitates the spatialization process. Thus, as an example, spatialization was performed using the CP and PF packages (Figures S10 and S11) with the MCD13 product. The quickest algorithm was the GB, and the slowest was the TM (Table 2).

**Table 2.** Runtime for 100 NDVI time series for each package, with CP (CropPhenology), DT (DEA tools), GB (greenbrown), PF (phenofit), PX (phenex), and TM (TIMESAT).

|          | CP    | DT    | GB    | PF    | PX     | TM     |
|----------|-------|-------|-------|-------|--------|--------|
| Time (s) | 0.343 | 6.879 | 0.130 | 0.146 | 10.278 | 76.544 |

These tests were performed in an Intel i5-7200U (7th Generation) with 8 GB RAM.

## 4. Discussion

Overall, we found that PX and TM were the best algorithms for phenology estimation, with the lowest errors (Table S1 and Figure 5). The algorithms had different performances when predicting SD or HD; we found that the TM algorithm was more suitable to predict sowing while PX was to predict the harvest. Packages based on the threshold method (such as PX and TM) showed higher consistency, except for GB (Figures S3 and S4). Large discrepancies were observed between the algorithms based on the derivative method (PF, GB, and DT using faster growth and senescence rate). Due to diverse methodologies, these inconsistencies are a source of uncertainty in detecting LSP [4,61]. Thus, even the best-performing algorithms lack the precision to drive in-season decisions (best performances with RMSE between 10 and 20 days—Table S1). Still, these methods can provide trends and assess regional practice changes much quicker than surveys. For example, detecting sowing practices changes with the introduction of new crop cultivars, i.e., a cultivar that could be sown deeper and, hence, earlier in dry lands to capture early rains [62] or crop models that could be informed to evaluate large regions [63].

Although small differences were found, the remote sensing data sources that showed the best results were Sentinel-2 and MCD13 for most packages (Table S1 and Figure 5). Studies that have analyzed different data sources have observed that estimates of phenological metrics could be affected when the temporal [64] and spatial resolution [65] of the sensor changed. However, here, the TM package was the most robust and showed similar behavior and results regardless of the data sources (Table S1 and Figure 7a,b). On the other side, GB and DT seem to be more sensitive to the data source used (Figure 7c,d).

The use of different VIs had a stronger impact on the phenology extraction performance than the changing spatial and temporal resolution (Figure 4). This is probably related to the fact that EVI and EVI2 were developed to overcome problems with saturation, minimize atmospheric contamination, and reduce the bare-ground signal present in NDVI [44,66,67].

Estimating phenological metrics in agricultural regions with more than one cycle makes this process even more challenging [68,69]. Double cropping, a common practice in the studied regions (soy and corn in the south region [40] and soy, corn, cotton, or non-commercial crops in the north [70]), generates confusion in the algorithms (Figure 8) because of the spectral pattern similarity of these crops [70]. Although TM has features that allow to identify the number of the crop cycle and set the desired cycle [71], it often ends up selecting the wrong cycle regardless—particularly harvest dates (Figure 8b). This problem could be solved by inserting the time series corresponding exactly to the period of soybeans grown in the region. However, this information is not available, and it would require a classification step that would make estimates even more challenging for large regions because soybeans could be the first or the second crop in the season.

In addition, the validation of phenological metrics estimates for agricultural areas is still difficult because of the gap between remotely sensed data and those observed in the field [17,72]. That is, the sowing date differs from the emergence date, where only the latter will change the VI values. Hence, although no method explicitly accounts for it, when using VIs to identify sowing and harvest dates, the difference between sowing and crop emergence needs to be accounted for. Note that setting a fixed number of days between sowing and emergence needs to be inferred based on crop- and region-specific assumptions [72]. Furthermore, different philosophies—not based on Vis—were explored. For example, [17] used ultra-high spatial and temporal resolution satellites and obtained an RMSE of two days. However, this method is more computationally intensive than VIs-based methods and was evaluated on a small scale. In addition, other works also performed a re-scaling of SOS to obtain sowing dates by subtracting a value in days and showed the potential of this methodology [73,74].

Nevertheless, the limitations and uncertainties could potentially be reduced with methods and data sources that were not tested in this study. For example, calibrating each algorithm for a specific region with a similar growing pattern could improve the accuracy of the estimates. Bias removal methods, such as those proposed by [75] and [76], could be used to increase the reliability of the data and their use to guide management practices. Another opportunity that could improve this type of study is the synthetic aperture radar (SAR) data [77], which does not suffer from cloud interference [55]. Also, data sources with a finer spatial and temporal resolution, such as the Harmonized Landsat and Sentinel-2 (HLS) project [78] and the cube-satellite constellations (e.g., PlanetScope) [79], can provide more accurate time series. We have not tested these different data sources in this study; we know about their potential, but more research needs to be conducted since radar backscattering has presented a low correlation with the observed data so far [55]. Furthermore, PlanetScope images tend to suffer from inter-sensor inconsistencies that introduce noise into the time series of observations [80], and by the time we concluded this study, the HLS was not available in GEE. Nonetheless, this study provides insights when using well-documented methods to estimate phonology using vegetation indices from diverse sources.

Furthermore, works that had higher correspondence with field data used small areas and small validation datasets. For example, 50 fields for [17] and 100 fields for [55] when compared with this study, where more than 900 fields were used for validation, distributed across large production areas of Brazil (with fields more than 2000 km apart). Thus, this work highlights that after twenty years of investigation, new approaches must consider the variation in different ecosystems and the easy application over large areas. This would support phenology investigations in moving from examining past phenology to predicting future phenology [37,81].

## 5. Conclusions

TIMESAT and phenex presented the most accurate estimates of phenological metrics for sowing and harvest dates, respectively. TIMESAT was less sensitive to changes in spatial, temporal, spectral, and radiometric changes, while the other methods were sensitive to changes in data input. Overall, after two decades of development, sowing and harvest dates extracted from satellite vegetation indices errors are between 10 and 20 days. Thus, its applicability is limited for precise and localized decision-making; regional assessments that can cope with such errors benefit from these methods instead of surveys, nonetheless. More studies using different algorithms and data sources can help to improve the accuracy of estimates. Despite the limitations and uncertainties, this study provides a base for future studies that can lead to new approaches and improvements in the algorithms developed so far.

**Supplementary Materials:** The following supporting information can be downloaded at https://www.mdpi.com/article/10.3390/rs15225366/s1; Section S1: Smoothing time series tests; Section S2: Parameter calibration; Figure S3: Correlation coefficient between the data observed in the field and SD and HD estimates for the different parameters of each tested. CP (CropPhenology), DT (DEA tools), PF (phenofit), PX (Phenex), and TM (TIMESAT); Figure S4: Mean difference and standard deviation between the observed SD (left) and HD (right) field data and the different estimates made using different parameters for the packages. CP (CropPhenology), DT (DEA tools), PF (phenofit), PX (Phenex), and TM (TIMESAT); Figure S5: Distribution of each tested dataset. Black lines indicate the first and third quantiles. Red lines indicate the quantiles from field data; Figure S6: Spatial pattern of the difference between observed and estimated sowing date (a–f) and harvest dates (g–l) estimated by the different packages tested for MCD13 EVI2, with CropPhenology (a,g), DEA tools (b,h), greenbrown (c,i), phenofit (d,j), Phenex (e,k), and TIMESAT (f,l). Root meant square error (RMSE) is in days; Figure S7: Spatial pattern of the difference between observed and estimated sowing date (a–f) and harvest dates (g–l) estimated by the different packages tested for MOD09 EVI2, with CropPhenology (a,g), DEA tools (b,h), greenbrown (c,i), phenofit (d,j), Phenex (e,k), and TIMESAT (f,l). Root meant square error (RMSE) is in days; Figure S8: Spatial pattern of the difference between observed and estimated sowing date (a–f) and harvest dates (g–l) estimated by the different packages tested for MOD13 EVI2, with CropPhenology (a,g), DEA tools (b,h), greenbrown (c,i), phenofit (d,j), Phenex (e,k), and TIMESAT (f,l). Root meant square error (RMSE) is in days; Figure S9: Spatial pattern of the difference between observed and estimated sowing date (a–f) and harvest dates (g–l) estimated by the different packages tested for Sentinel-2 EVI2, with CropPhenology (a, g), DEA tools (b,h), greenbrown (c,i), phenofit (d,j), Phenex (e,k), and TIMESAT (f,l). Root meant square error (RMSE) is in days; Figure S10: Sowing and harvest dates estimates made using the CropPhenology (T_CP, threshold method) and phenofit (D_PF, derivative method) packages; Figure S11: Cycle duration estimated using the CropPhenology (T_CP, threshold method) and phenofit (D_PF, derivative method) packages; Table S1: Correlation coefficient, RMSE (days), bias (%), and standard deviation (days) for all estimates, with CP (CropPhenology), DT (DEA tools), GB (greenbrown), PF (phenofit), PX (Phenex), and TM (TIMESAT).

**Author Contributions:** Conceptualization, G.R., I.D.S. and J.R.; methodology, G.R., I.D.S., J.R. and M.A.; software, G.R. and H.d.N.B.; validation, G.R., I.D.S., J.R., R.Y.T., R.L., H.d.N.B. and M.A.; formal analysis, G.R., I.D.S., J.R., R.Y.T., R.L., H.d.N.B. and M.A.; investigation, G.R., I.D.S., J.R., R.Y.T., R.L., H.d.N.B. and M.A.; resources, G.R., I.D.S. and R.Y.T.; data curation, G.R., I.D.S., J.R, R.Y.T. and M.A.; writing—original draft preparation, G.R.; writing—review and editing, G.R., I.D.S., J.R., R.Y.T., R.L., H.d.N.B. and M.A.; visualization, G.R., I.D.S., J.R., R.Y.T., R.L., H.d.N.B. and M.A.; supervision, I.D.S. and J.R.; project administration, I.D.S. and J.R.; funding acquisition, G.R., I.D.S. and M.A. All authors have read and agreed to the published version of the manuscript.

**Funding:** This study was financed in part by the Coordenação de Aperfeiçoamento de Pessoal de Nível Superior (CAPES), Brazil; Finance Code 001. Special thanks to Fundação ABC for providing the field data. The authors are grateful to the Brazilian National Council of Scientific and Technological Development (CNPq) for the PhD grant of Rodigheri, G. [grant number 141410/2020-5] and Research Productivity Fellowship of Sanches, I.D [310042/2021-6] and Adami, M. [PQ 306334/2020-8].

**Data Availability Statement:** More information and reproducible code are available via GitHub and can be accessed using the following link: https://github.com/grazirodigheri/get-phenometrics. The field data are still private property and, therefore, could not be made available.

**Acknowledgments:** Special thanks to the Fundação ABC for the partnership in the experimental area of collecting and sharing the field data.

**Conflicts of Interest:** The authors declare no conflict of interest.

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
