# Peer review of "Estimating Crop Sowing and Harvesting Dates Using Satellite Vegetation Index: A Comparative Analysis"

_remotesensing, doi:10.3390/rs15225366_

Round 1

Reviewer 1 Report

Comments and Suggestions for Authors

General:

My congratulations to the authors crafting such timely and relevant paper using remote sensing. 

Study novelty needs to be highlighted more especially after identifying the problem statement which is the lack of LSP analysis using ground data and inter-comparison using different algorithms aided by remote sensing data

I admire the nice logical figures of the authors and the fact that such inter-comparison study would involve technical expertise in implementing different algorithms.

I find the study not paying too much attention about “uncertainty”. The authors could have contextualized the uncertainty component better given they have results concerning such e.g. Figure 4

Figure 5 shows common results. There is over estimation of low and underestimation of high number of days. It would be better to include in the discussion how to deal with such systematic errors/biases. Consider referring to bias removal methods e.g.

·         Precipitation Trends Over Mainland China From 1961–2016 After Removal of Measurement Biases - Zhang - 2020 - Journal of Geophysical Research: Atmospheres - Wiley Online Library

·         A comprehensive framework for assessing the accuracy and uncertainty of global above-ground biomass maps - ScienceDirect

Potential systematic error get worse looking at Figure 8 right figure, which made me think whether such is a calibration issue?

A find the method relatively long with texts of less importance. Consider annexing some texts like e.g. long pre-processing of satellite data

The main result is the limitation of the LSE algorithms in general but I don’t see further discussion on ways to move forward. I have a feeling using too much optical data in a tropical environment for a study period of 2 years would mean something. Did the authors consider Sentinel 1 backscatter? Better to elaborate further on other uncertainty sources could have influenced the main finding?

I crave for more recommendations out of the findings. Can you specifically suggest wyas to move forward? Are the models worth “ensembling”? Can the systematic errors be removed / models can be re-calibrated? Can local drone data be integrated? Can Planet 3m be used? Dynamic Landsat-based datasets e.g. Potapov and Hansen datasets be integrated? Let us know…

I feel that the authors are being too conclusive despite the own limitations of their experiment i.e. using 2 years of data and without proper “uncertainty analysis” and even sensitivity analysis i.e. trying different parameter combinations of LSP models. I don’t want to have an impression that the authors get to personal with the science of remote sensing-assisted LSP. This thought is also reflected the way the title is presented. I suggest revising the title in that regard.

Specific:

Line 18 “implemented in different algorithms” a bit vague

Define DEA first

Have you seen related studies that did inter-comparisons of LSP algorithms even for other commodities?

Avoid enclosing in parenthesis as much as possible. Texts inside parenthesis are deemed less important.

Include in the Abstract the study area

Line 43 too much citations! And weird that the second statement has one.

Line 45 why challenging?

Just curious, are the soy plantations somehow outcomes of deforestation? :-(

The Introduction lacks statements about the study novelty i.e. how does your approach using MODIS and Sentinel 2 would address the problem of having many phenology assessment tools especially the satellite data date and ground data spatial and temporal mismatches.

Figure 1 is nice!

The choice of the vegetation indices need to be motivated more

Can you say more about how the models guideline about large scale predictions?

Can you say something about the ground data sample? Are they based on probability samples e.g. random?

Can you also say something about the gaps e.g. due to cloud cover, I’m a bit worried how much has been interpolated. Note that you have 2 years of temporal study period. This remind me that the authors should highlight the study period more.

Figure 4 is cluttered and not eye friendly.

The authors may need to discuss their own (study) limitation.

Comments on the Quality of English Language

Good English in general. 

Author Response

Dear Reviewer,

Please see the file attached.

Thank you so much for your revision.

Regards,

Reviewer 2 Report

Comments and Suggestions for Authors

The article contribute to the Remote Sensing journal, however is important to add some comments that I already did to the article body.

In all article body please homogenize 8-day or 8 day.

Also homogenize 2 annual or two annual, 3 algorithms or three algorithms.

In Figures 5 and 8 please write the black line represents the 1:1 line.

Author Response

(The authors gave the same response as above.)

Reviewer 3 Report

Comments and Suggestions for Authors

Overall Impression
In this manuscript, the authors compare the sensitivity and efficacy of six phenological tools for assigning sowing and harvest dates for soybeans in Brazil, using three remotely sensed data sources. The authors find that one of the older methods, TIMESAT, remains the most robust to both data type and quality, and that current tools are unable to inform in-season decision-making.

The manuscript is generally well-written (see line-by-line comments for a non-exhaustive list of minor typos and grammar issues), with methods and data described well. The results are somewhat difficult to parse through, due to the sheer number of tools and datasets producing an alphabet soup, so to speak, of acronyms. The figures also suffer from this, making it difficult for me as a non-expert in crop phenology to quickly interpret the results. However, the results and discussion seem to derive logically from the methods and data, and I am generally satisfied that this manuscript offers a solid comparison of the tools assessed.

If the authors address the comments below, I would be happy to recommend the manuscript for publication in Remote Sensing.

General Comments
The figures are very dense, to the point of overwhelming (e.g., the twelve panels of Figure 6). I’m not sure what could be done to simplify them, but perhaps a rearrangement or more assertive panel titling could help.

Multiple times, “emergency date” seems to be used instead of “emergence date”. I suggest changing all these to the latter.

Line by Line Comments
103: “To understand how the algorithm’s robustness are” would be clearer if restructured to “To understand how robust the algorithm are”

214: “in R language” should be “in the R language”

246: “parameters” instead of “parameter”

313-315: There are a couple strange errors or spellings, which make me think part of the intended sentence was omitted.

382: “may difficult the estimates” needs rewording.

Comments on the Quality of English Language

See above comments.

Author Response

(The authors gave the same response as above.)

Round 2

Reviewer 1 Report

Comments and Suggestions for Authors

Github link not working, pls resolve before publication